# AN OPTIMIZATION PRINCIPLE OF DEEP LEARNING?

## ABSTRACT

Training deep neural networks (DNNs) has achieved great success in recent years. Modern DNN trainings utilize various types of training techniques that are developed in different aspects, e.g., activation functions for neurons, batch normalization for hidden layers, skip connections for network architecture and stochastic algorithms for optimization. Despite the effectiveness of these techniques, it is still mysterious how they help accelerate DNN trainings in practice. In this paper, we propose an optimization principle that is parameterized by $\gamma > 0$ for stochastic algorithms in nonconvex and over-parameterized optimization. The principle guarantees the convergence of stochastic algorithms to a global minimum with a monotonically diminishing parameter distance to the minimizer and leads to a $\mathcal{O}(1/\gamma K)$ sub-linear convergence rate, where $K$ is the number of iterations. Through extensive experiments, we show that DNN trainings consistently obey the $\gamma$-optimization principle and its theoretical implications. In particular, we observe that the trainings that apply the training techniques achieve accelerated convergence and obey the principle with a large $\gamma$, which is consistent with the $\mathcal{O}(1/\gamma K)$ convergence rate result under the optimization principle. We think the $\gamma$-optimization principle captures and quantifies the impacts of various DNN training techniques and can be of independent interest from a theoretical perspective.

## 1 INTRODUCTION

Deep learning has been successfully applied to various domains such as computer vision, natural language processing, etc, and has achieved state-of-art performance in solving challenging tasks. Although deep neural networks (DNNs) have been well-known for decades to have great expressive power Cybenko (1989), the empirical success of training DNNs postponed to recent years when sufficient computation power is accessible and effective DNN *training techniques* are developed.

The milestone developments of DNN training techniques can be divided into two categories. First, various techniques have been developed at different levels of *neural network design*. At the neuron level, various functions have been applied to activate the neurons, e.g., sigmoid function, hyperbolic tangent (tanh) function and the more popular rectified linear unit (ReLU) function Nair & Hinton (2010). At the layer level, batch normalization (BN) has been widely applied to the hidden layers of DNNs to stabilize the training Ioffe & Szegedy (2015). Moreover, at the architecture level, skip connections have been introduced to enable successful training of deep networks He et al. (2015); Szegedy et al. (2015); Srivastava et al. (2015); Huang et al. (2016). Second, various efficient stochastic *optimization algorithms* have been developed for DNN training, e.g., stochastic gradient descent (SGD) Robbins & Monro (1951); Rumelhart et al. (1986), SGD with momentum Qian (1999); Nesterov (2014) and Adam Kingma & Ba (2015), etc. Table 1 provides a summary of these important DNN training techniques.

Table 1: Summary of DNN training techniques

| Neuron activation | Layer normalization | Network architecture | Optimization algorithm |
|---|---|---|---|
| sigmoid, tanh, ReLU | batch normalization | skip connection | SGD, SGD-momemtum, Adam |

While these training techniques have been widely applied in practical DNN training, there is limited understanding on how they help facilitate the training and achieve the global minimum of the network. In the existing literature, it is known that the sigmoid and tanh activation functions can cause the vanishing gradient problem, and the ReLU activation function is a popular replacement that avoids this issue Nair & Hinton (2010); Glorot et al. (2011). On the other hand, the batch normalization is originally proposed to reduce the internal covariance shift Ioffe & Szegedy (2015), and more recent studies show that it allows to use a large learning rate Bjorck et al. (2018) and improves the loss landscape Santurkar et al. (2018). The skip connection has been shown to help eliminate singularities and degeneracies Orhan & Pitkow (2018) and improve the loss landscape Hardt & Ma (2016). Moreover, regarding the optimization algorithm, the momentum scheme has been well-known to accelerate convex optimization Nesterov (2014) and is also widely applied to accelerate nonconvex optimization Ghadimi & Lan (2016), whereas the Adam algorithm normalizes the update in each dimension to accelerate deep learning Kingma & Ba (2015). While these studies provide partial explanations to the effectiveness of various DNN training techniques, their reasonings are from very different perspectives and it is unclear whether a general principle can exist that guides these training techniques to facilitate the training process. In particular, these studies do not explain why DNN trainings with different techniques can achieve the global minimum in practice. Moreover, the existing explanations cannot quantify the impacts of the training techniques on training deep networks, and a principled quantification metric is still far from clear.

This paper attempts to propose an optimization principle that characterizes and quantifies the impacts of various training techniques on DNN training. The proposed principle is applicable to general stochastic algorithms in the nonconvex and over-parameterized regime and leads to guaranteed convergence to a global minimum. We summarize our contributions as follows.

## 1.1 OUR CONTRIBUTIONS

**Theory:** We propose an optimization principle that is parameterized by $\gamma > 0$ for stochastic algorithms in nonconvex and over-parameterized optimization. The $\gamma$-optimization principle (see Definition 1) requires the current update points toward a global minimizer, and the inner product between both directions is lower-bounded by the update norm and the loss gap that is scaled by the factor $\gamma$. We show that the $\gamma$-optimization principle guarantees the optimization path generated by stochastic algorithms to approach a global minimizer with a monotonically diminishing distance. Moreover, it leads to convergence of the optimization path to the global minimum with a $\mathcal{O}(1/\gamma K)$ sub-linear convergence rate .

**Experiments:** We conduct extensive experiments to examine the validity of the $\gamma$-optimization principle in DNN training. In specific, we find that all the tested DNN trainings obey the $\gamma$-optimization principle throughout the training process, and the generated optimization paths satisfy the theoretical properties of the principle that are mentioned above. Moreover, we explore the impacts of the training techniques that are listed in Table 1 on the parameterization $\gamma$ of the optimization principle. We find that DNN trainings that apply the training techniques achieve fast convergence and obey the optimization principle with a large $\gamma$. As a comparison, DNN trainings without the training techniques converge slowly and obey the optimization principle with a small $\gamma$. These observations are consistent with the convergence rate characterization of the $\gamma$-optimization principle that a larger $\gamma$ leads to faster convergence. Therefore, the $\gamma$-optimization principle captures and quantifies the impacts of these training techniques on DNN training in a unified way through the parameter $\gamma$, and sheds light on the underlying mechanism that leads to successful DNN training.

## 1.2 RELATED WORK

**DNN training techniques:** Various training techniques have been developed for DNN training. Examples include piece-wise linear activation functions, e.g., ReLU Nair & Hinton (2010), ELU Clevert et al. (2015), leaky ReLU Maas et al. (2013), batch normalization Ioffe & Szegedy (2015), skip connection He et al. (2015); Szegedy et al. (2015); Srivastava et al. (2015) and advanced optimizers such as SGD with momentum Qian (1999), Adagrad Duchi et al. (2011), Adam Kingma & Ba (2015), AMSgrad Reddi et al. (2018). The ReLU activation function and skip connection have been shown to help avoid the vanishing gradient problem and improve the loss landscape Hardt & Ma (2016); Zhou & Liang (2017); Zhang et al. (2019); Zou et al. (2018). The batch normalization has been shown to help avoid the internal covariance shift problem Ioffe & Szegedy (2015). More recent studies show that batch normalization allows to adopt a large learning rate Bjorck et al. (2018)

and improves the loss landscape in the training Santurkar et al. (2018). The convergence properties of the advanced optimizers have been studied in nonconvex optimization Chen et al. (2019).

**Optimization properties of nonconvex ML:** Many nonconvex ML models have amenable properties for accelerating the optimization. For example, nonconvex problems such as phase retrieval Zhang et al. (2017b), low-rank matrix recovery Tu et al. (2016), blind deconvolution Li et al. (2018b) and neural network sensing Zhong et al. (2017) satisfy the local regularity geometry around the global minimum Zhou et al. (2016); Tu et al. (2016); Li et al. (2018b); Zhong et al. (2017); Zhou & Liang (2017), which guarantees the linear convergence of gradient-based algorithms. More recently, DNN trainings that use SGD have been shown to follow a star-convex optimization path Zhou et al. (2019).

## 2 AN OPTIMIZATION PRINCIPLE FOR OVER-PARAMETERIZED ML

In this section, we introduce the $\gamma$-optimization principle that provides convergence guarantees for over-parameterized nonconvex machine learning (ML). We show empirically in the subsequent sections that practical DNN trainings obey this principle.

### 2.1 GLOBAL MINIMUM OF OVER-PARAMETERIZED MODELS

The goal of a machine learning task is to search for a good ML model $\theta$ that minimizes the total loss $f$ on a set of training data samples $\mathcal{Z} : \{z_i\}_{i=1}^n$. The problem is formally written as

$$\min_{\theta \in \mathbb{R}^d} f(\theta; \mathcal{Z}) := \frac{1}{n} \sum_{i=1}^n \ell(\theta; z_i), \tag{P}$$

where $\ell(\cdot; z_i) : \mathbb{R}^d \to \mathbb{R}$ corresponds to the loss on the $i$-th data sample $z_i$. For many nonconvex ML models, e.g., deep neural networks, a prominent feature is the over-parameterization of the model, i.e., the model capacity is sufficient to over-fit all the training data samples. In another word, the global minimizers of the total loss $f$ under an over-parameterized model are common minimizers of all the individual loss functions $\ell(\cdot; z_i), i = 1, ..., n$. We summarize this fact formally as follows.

**Fact 1.** *Consider the problem (P) with an over-parameterized model $\theta$. Every global minimizer $\theta^*$ of the total loss $f$ is also a global minimizer of each individual loss $\ell(\cdot; z_i)$ for all $i = 1, ..., n$.*

To elaborate, note that we typically use non-negative loss in DNN training. Therefore, if the total loss $f$ is trained to achieve zero (i.e., the global minimum), then each individual loss $\ell(\cdot; z_i)$ must also achieve zero and the obtained model over-fits the training data. Of course, in practice, due to the termination of the optimization process within finite number of iterations, the total loss can only achieve an approximate global minimum that is very close to zero. Throughout the rest of this section, we assume that Fact 1 holds for the problem (P).

### 2.2 THE $\gamma$-OPTIMIZATION PRINCIPLE

Consider a generic stochastic algorithm (SA) that is initialized with certain model $\theta_0$. In each iteration $k$, the SA samples a data sample $z_{\xi_k} \in \mathcal{Z}$, where $\xi_k \in \{1, ..., n\}$ is obtained via cyclic sampling with reshuffle. Based on the current model $\theta_k$ and the sampled data $z_{\xi_k}$, the SA generates a stochastic update $U(\theta_k; z_{\xi_k})$ and applies it to update the model with a learning rate $\eta > 0$ according to

$$(\text{SA}): \quad \theta_{k+1} = \theta_k - \eta U(\theta_k; z_{\xi_k}), \quad k = 0, 1, 2, ... \tag{1}$$

Equation (1) covers the update rule of many existing optimizers for DNN training. For example, the stochastic gradient descent (SGD) algorithm chooses the update $U(\theta_k; z_{\xi_k})$ to be the stochastic gradient $\nabla_\theta \ell(\theta_k; z_{\xi_k})$. In comparison, the SGD with momentum algorithm generates the update using an extra momentum step, and the Adam algorithm generates the update as a moving average of the stochastic gradients normalized by the moving average of their second moments [1].

We next introduce the $\gamma$-optimization principle for the SA.

**Definition 1** ($\gamma$-optimization principle for SA)**.** *Apply SA to solve the over-parameterized problem (P). For a certain global minimizer $\theta^*$ of the problem (P), the optimization path $\{\theta_k\}_k$ generated by the SA satisfies: for certain $\gamma > 0$ and all $k = 0, 1, 2, ...,$,*

$$\langle \theta_k - \theta^*, U(\theta_k; z_{\xi_k}) \rangle \geq \frac{\eta}{2} \|U(\theta_k; z_{\xi_k})\|^2 + \gamma \|\theta_0 - \theta^*\|^2 \big( \ell(\theta_k; z_{\xi_k}) - \ell(\theta^*; z_{\xi_k}) \big). \tag{2}$$

---

[1]The Adam updates depend on the past stochastic samples.

To elaborate, the left hand side of eq. (2) measures the coherence between $\theta_k - \theta^*$ and the current model update $U(\theta_k; z_{\xi_k})$. Intuitively, a positive coherence implies that the current model update points toward the global minimizer $\theta^*$ from the current model $\theta_k$. On the other hand, the right hand side of eq. (2) regularizes the coherence by two non-negative terms: the square norm of the model update scaled by the learning rate $\eta$ and the optimality gap of the loss on the sampled data scaled by the constant $\gamma \|\theta_0 - \theta^*\|^2$. Hence, a larger value of $\gamma$ implies that the update $U(\theta_k; z_{\xi_k})$ is more coherent with the direction towards the minimizer $\theta_k - \theta^*$ and therefore facilitates the convergence. We show in Theorem 1 later that the $\gamma$ plays a central role in characterizing the convergence rate of the SA.

**Discussion:** The $\gamma$-optimization principle is related to other existing ones in nonconvex ML. In specific, nonconvex problems such as phase retrieval Zhang et al. (2017b) and low-rank matrix recovery Tu et al. (2016) have been shown to satisfy the regularity condition, which replaces the last term in eq. (2) by $\gamma \|\theta_k - \theta^*\|^2$ and guarantees the linear convergence of $\{\theta_k\}_k$ to $\theta^*$. However, such a fast convergence rate does not hold in practical DNN training. On the other hand, DNN trainings have been shown in Zhou et al. (2019) to follow a star-convex optimization path, which corresponds to eq. (2) with $\gamma = \|\theta_0 - \theta^*\|^{-2}$, $U(\theta_k; z_{\xi_k}) = \nabla \ell(\theta_k; z_{\xi_k})$ and absence of the term $\frac{\eta}{2}\|U(\theta_k; z_{\xi_k})\|^2$. However, such a principle is applicable to SGD updates only and an additional global Lipschitz assumption on the loss function is required in order to have theoretical convergence guarantee. In comparison, the $\gamma$-optimization principle is applicable to the more general SA and does not rely on additional assumption to have convergence guarantee as we elaborate below.

We obtain the following convergence results of SA under the $\gamma$-optimization principle.

**Theorem 1** (Convergence of SA). *Apply SA to solve the over-parameterized problem (P). If the optimization process satisfies the principle in Definition 1, then the following statements hold.*

1. *The optimization path $\{\theta_k\}_k$ approaches the global minimizer $\theta^*$ with a monotonically diminishing distance, i.e., $\|\theta_{k+1} - \theta^*\| \leq \|\theta_k - \theta^*\|$ for all $k = 0, 1, 2, ....,$.*

2. *For all $i$, each $\ell(\cdot; z_i)$ converges to its global minimum along the optimization path, i.e., denote $\{i(T)\}_{T \in \mathbb{N}}$ as the sequence of iterations that sample $z_i$, then, $\lim_{T \to \infty} \ell(\theta_{i(T)}; z_i) = \ell(\theta^*; z_i)$.*

3. *For any $K = nB, B \in \mathbb{N}$, the accumulated loss converges to the global minimum at the rate*

$$\frac{1}{K} \sum_{k=0}^{K-1} \ell(\theta_k; z_{\xi_k}) - f(\theta^*; \mathcal{Z}) \leq \frac{1}{2\eta \gamma K}. \tag{3}$$

To elaborate, item 1 of Theorem 1 shows that the optimization path of SA approaches the global minimizer with a monotonically diminishing distance under the $\gamma$-optimization principle, and item 2 further establishes the convergence of each individual loss to its global minimum. More importantly, item 3 shows that the average of the accumulated loss convergences to the global minimum of the total loss at a sub-linear convergence rate, which is inverse proportionally to the parameterization $\gamma$ of the optimization principle in eq. (2). Intuitively, a larger $\gamma$ implies a more coherent update $U(\theta_k; z_{\xi_k})$ with the desired direction $\theta_k - \theta^*$ and therefore leads to a faster convergence.

**Remark:** Under the $\gamma$-optimization principle, the sub-linear convergence rate in item 3 depends on the learning rate $\eta$ and the parameter $\gamma$ only, which are problem-independent parameters. Therefore, given a fixed learning rate, the parameter $\gamma$ of the optimization principle provides a universal quantification of the optimization quality. Inspired by this idea, in the subsequent sections, we conduct extensive experiments to examine the validity of the $\gamma$-optimization principle in DNN training. In particular, we empirically quantify the impacts of different training techniques on the DNN training by evaluating the parameter $\gamma$ of the optimization principle in each optimization process.

## 3 EXPERIMENTS ON NETWORK-LEVEL TRAINING TECHNIQUES

In this section, we examine the validity of the $\gamma$-optimization principle in training DNNs with different neural network-level training techniques, i.e., activation function, batch normalization and skip connection. We outline the exploration plan below and provide the details of the experiment setup in the corresponding subsections.

**Exploration plan:** In all DNN trainings, we train the network for a sufficient number of epochs to achieve an approximate global minimum. We store the network parameters $\{\theta_k\}_k$, loss $\{\ell(\theta_k; z_{\xi_k})\}_k$ and update $\{U(\theta_k; z_{\xi_k})\}_k$ that are generated in each DNN training. Then, we compute the upper bound for $\gamma$ in each iteration according to eq. (2), where we set $\theta^*$ to be the network parameters produced in the last training iteration. We use mini-batch sampling and all the stored loss and updates correspond to their mini-batch versions.

### 3.1 IMPACT OF ACTIVATION FUNCTION ON $\gamma$-OPTIMIZATION PRINCIPLE

**Experiment setup:** We train a variant of the Alexnet Zhang et al. (2017a) and the Resnet-18 He et al. (2015) with different choices of activation functions for all the nonlinear neurons. The activation functions that we explore include sigmoid, tanh, ReLU and leaky ReLU (with slope $10^{-2}$). We apply the standard SGD optimizer with a fixed initialization point, a mini-batch size 128 and a constant learning rate $\eta = 0.05$ to train these networks for 150 epochs on the Cifar10 and Cifar100 datasets Krizhevsky (2009), respectively.

In the three rows of Figure 1, we present the training results of the normalized distance-to-minimizer, training loss and parameter $\gamma$ of the optimization principle along the optimization path, respectively. From the figures in the third row, one can see that these trainings with different activation functions obey the $\gamma$-optimization principle with $\gamma > 0$ in the training process. This demonstrates the validity of the principle in these DNN trainings. Also, the figures in the first and second rows respectively show that the distance-to-minimizer diminishes monotonically and the training loss converges to the global minimum in these trainings. These observations are consistent with the theoretical implications of the $\gamma$-optimization principle in items 1 and 2 of Theorem 1.

Moreover, regarding the trainings of the Alexnet (first two columns), we observe that the trainings with the sigmoid activation function converge much slower than those with the other activation functions, and the parameter $\gamma$ under the sigmoid activation has a smaller value than those under the other activation functions. On the other hand, in the trainings of the Resnet-18 (last two columns), we observe that the convergence speed of the trainings with ReLU types of activation functions is the fastest, and is followed by that of the trainings with tanh activation function and sigmoid activation function, respectively. Moreover, one can see that the $\gamma$ in the trainings with ReLU types of activation functions has the largest value, and is followed by that in the trainings with tanh activation function and sigmoid activation function, respectively. These observations are consistent with item 3 of Theorem 1, where a larger $\gamma$ implies faster convergence.

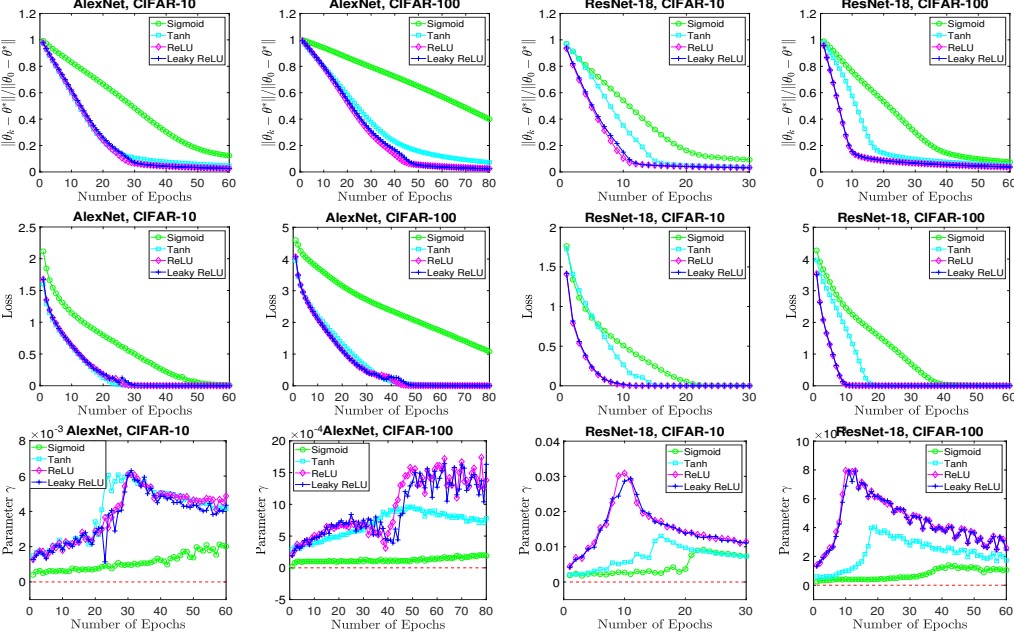

Figure 1: Training Alexnet and Resnet-18 with different activation functions.

The experiments in this subsection show that DNN trainings with different activation functions obey the $\gamma$-optimization principle, which captures and quantifies the impact of choice of activation function on the training process via the parameter $\gamma$.

### 3.2 IMPACT OF BATCH NORMALIZATION ON $\gamma$-OPTIMIZATION PRINCIPLE

**Experiment setup:** We train the Resnet-18, 34 and the Vgg-11, 16 networks with the settings: 1) keep all the BN layers; 2) keep the first BN layer in each block; and 3) remove all the BN layers. We remove all the dropout layers in the Vgg networks. We apply SGD with a fixed initialization point, a constant learning rate ($\eta = 0.05$ for Resnet, 0.01 for Vgg) and batch size 128 to train these networks on the Cifar10 and Cifar100 datasets, respectively.

Figure 2 shows the training results on the Cifar10 dataset. Due to space limitation, we present the distance-to-minimizer results in Appendix B, where one can see that the distances are monotonically diminishing and therefore are consistent with item 1 of Theorem 1. In all the trainings shown in Figure 2, removing all BN layers significantly slows down the convergence of loss. In particular, vanishing gradient occurs after around 25 epochs in the trainings of resnets without any BN layer and we plot the corresponding curves up to the epoch when gradient vanishes. On the other hand, the trainings that keep the first BN layer in each block converge as fast as those with all BN layers, and they all converge to the global minimum.

Regarding the $\gamma$ values, they are all positive throughout these trainings, which demonstrates the validity of the $\gamma$-optimization principle. Moreover, the $\gamma$ in the trainings without any BN layer is considerably smaller than that in the trainings with either one BN layer or all BN layers. Such an observation is consistent with the convergence rate result in item 3 of Theorem 1, where a larger $\gamma$ implies faster convergence.

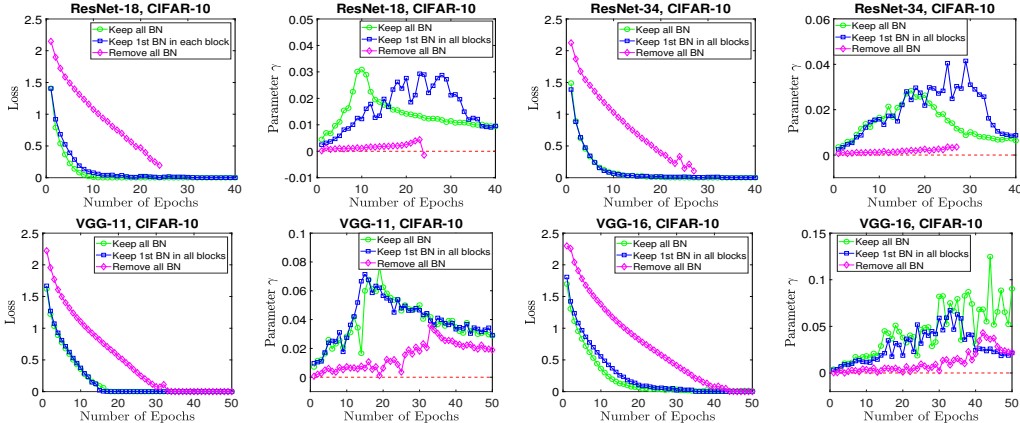

Figure 2: Training Resnets and Vggs with and without BN on Cifar10.

We also report in Figure 3 the results of training these networks on the Cifar100 dataset, where one can see the positiveness of the $\gamma$ throughout the training that demonstrates the validity of the $\gamma$-optimization principle. Particularly in the trainings of the resnets (see top row), the convergence of the trainings with all BN layers is the fastest, and is followed by that of the trainings with one BN layer in each block and without any BN layer, respectively. Moreover, in the first 20 epochs where the training loss is close to the global minimum, one can see that the $\gamma$ in the trainings with all BN layers is larger than that in the trainings with one BN layer in each block, and the $\gamma$ in the trainings without any BN layer has the smallest value. Similar observation is made in the training of the Vgg-16 network. Moreover, in the training of the Vgg-11 network, the training with one BN layer in each block is sufficient to achieve comparable performance to that of the training with all BN layers, and their $\gamma$ have similar values throughout the training. In comparison, the training without any BN layer is significantly slower and have much smaller $\gamma$ values.

The experiments in this subsection demonstrate that the $\gamma$-optimization principle is able to capture and quantify the impact of batch normalization on DNN training in terms of the parameter $\gamma$.

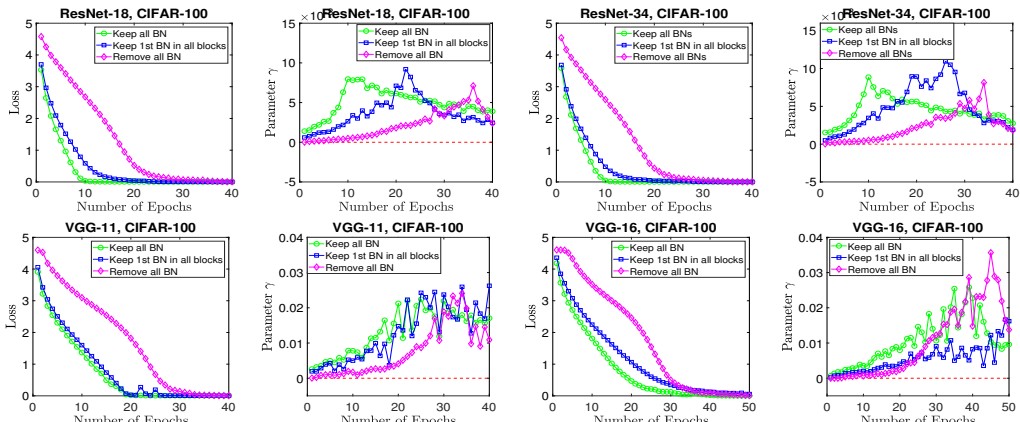

Figure 3: Training Resnets and Vggs with and without BN on Cifar100.

### 3.3 IMPACT OF SKIP CONNECTION ON $\gamma$-OPTIMIZATION PRINCIPLE

**Experiment setup:** We train the Resnet-18 with the settings: 1) keep all the skip connections; and 2) keep the first skip connection in each block. For the Resnet-34, we consider an additional setting where we keep the first two skip connections in each block. We apply SGD with a fixed initialization point, a constant learning rate $\eta = 0.05$ and batch size 128 to train these networks for 150 epochs on the Cifar10 and Cifar100 datasets, respectively.

Figure 4 shows the training results and we present the distance-to-minimizer results in Appendix C, where one can see that they are all monotonically diminishing in the trainings. One can see from the $\gamma$ curves that all these trainings satisfy the $\gamma$-optimization principle. In specific, regarding the trainings of the Resnet-18 on Cifar10, the training with one skip connection in each block has comparable performance to that of the training with all skip connections, and both trainings have comparable $\gamma$ values. In the trainings of the Resnet-18 on the more complex Cifar100 dataset, the training with more skip connections achieves a slightly faster convergence, and the corresponding $\gamma$ value is larger than that in the training with one skip connection in each block. Regarding the trainings of the deeper Resnet-34, it can be seen that skip connections significantly accelerate the training on both datasets. In particular, trainings are faster with more number of skip connections in each block, and the $\gamma$ values are larger in the trainings with more skip connections in each block. All these observations are consistent with the theoretical implications of the $\gamma$-optimization principle in Theorem 1. Therefore, the $\gamma$-optimization principle is able to characterize and quantify the impact of skip connections on DNN training. Our training results imply that skip connections are more effective in deeper resnets.

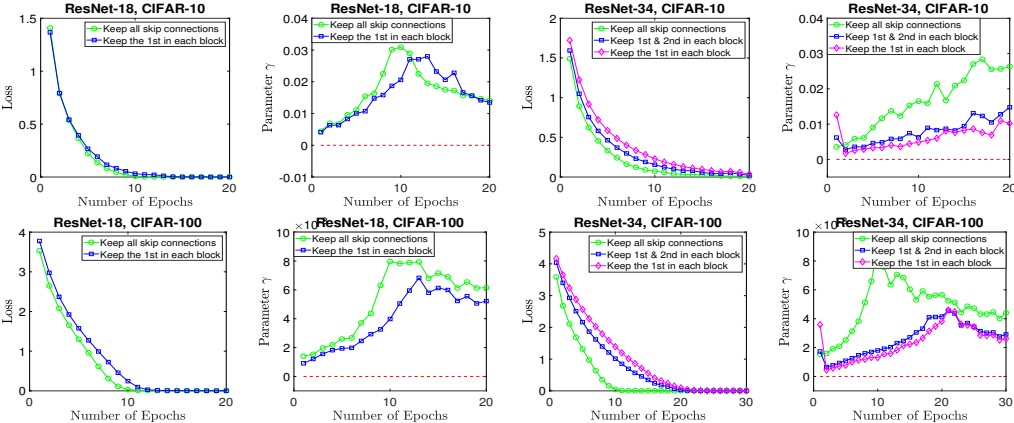

Figure 4: Training Resnets with and without skip connections.

## 4 EXPERIMENTS ON OPTIMIZATION-LEVEL TRAINING TECHNIQUES

In this section, we explore the impacts of stochastic optimizers on the $\gamma$-optimization principle.

**Experiment setup:** We train the Resnet-18, 34 and the Vgg-11, 16 networks using SGD, SGD with momentum and Adam, respectively. We remove all the dropout layers in the Vgg networks. We apply a fixed initialization point, a constant learning rate $\eta = 0.001$ and batch size 128 to all the optimizers and train these networks on the Cifar10 and Cifar100 datasets, respectively. We set the momentum to be 0.5 for the SGD with momentum and set $\beta_1 = 0.9, \beta_2 = 0.999, \epsilon = 10^{-2}$ for the Adam.

Figure 5 presents the training results on the Cifar10 dataset, and we present the distance-to-minimizer results in Appendix D.1 where one can see that they are all monotonically diminishing. In all these trainings, the $\gamma$ curves are above zero and therefore demonstrates the validity of the $\gamma$-optimization principle. Also, it can be seen that the SGD with momentum trainings achieve significantly faster convergence speed than that achieved by the SGD trainings, and the Adam trainings converge the fastest. Regarding the $\gamma$, it can be seen that the SGD trainings obey the optimization principle with the smallest $\gamma$, which is further enlarged in the SGD with momentum trainings. Moreover, the Adam trainings obey the optimization principle with the largest $\gamma$. These observations are consistent with item 3 of Theorem 1, where a larger $\gamma$ implies faster convergence of the training.

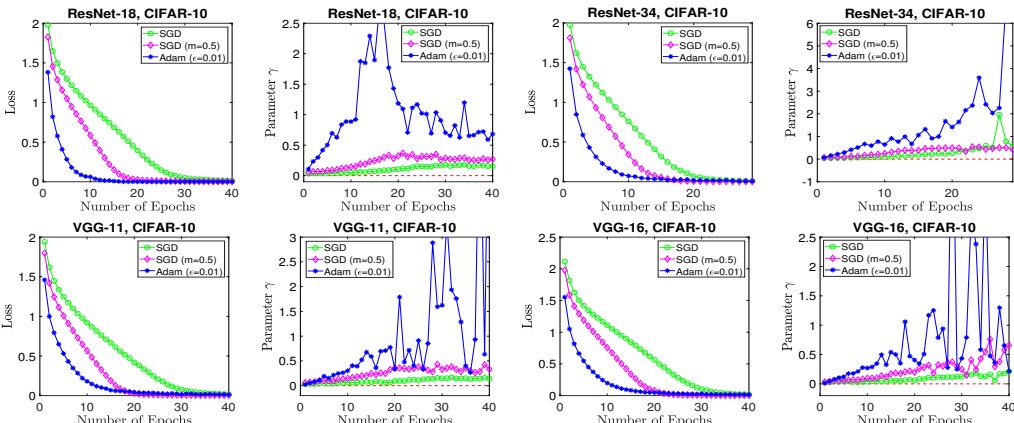

Figure 5: Training Resnets and Vggs with different optimizers on Cifar10.

The training results on the Cifar100 dataset are shown in Figure 6, where one can make very similar observations as those on Cifar10. In particular, for the Adam training of VGG-16 on Cifar100, the sign of $\gamma$ fluctuates after 50 epochs, which is possibly caused by the stochastic pre-conditioner in the Adam update. In addition to these experiments, we also visualize the optimization paths of different optimizers in Appendix D.2, where one can see that the directions of SGD updates are very different from those of the updates generated by the SGD with momentum and Adam. These experiments show that the $\gamma$-optimization principle characterizes the impact of optimizers on DNN training.

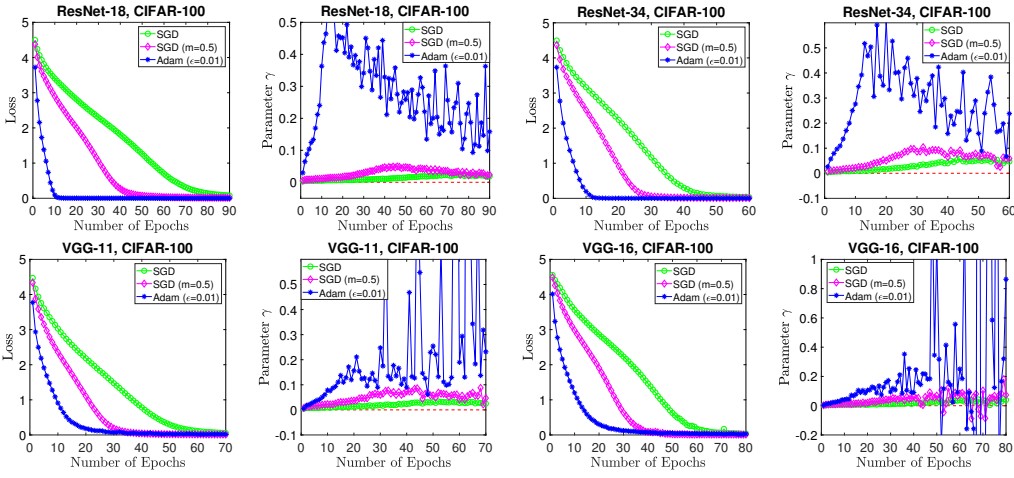

Figure 6: Training Resnets and Vggs with different optimizers on Cifar100.

## 5  CONCLUSION

In this paper, we propose a $\gamma$-optimization principle for general stochastic algorithms in nonconvex and over-parameterized optimization. The $\gamma$-optimization principle provides solid convergence guarantees to stochastic optimization and achieves a sub-linear convergence rate that scales inverse proportionally to $\gamma$ and is independent of the problem parameters. Through extensive DNN training experiments, we show that practical DNN trainings obey the principle reasonably well, and the parameter $\gamma$ provides a unified metric that measures the impacts of different training techniques on DNN training. In the future work, we expect that such an optimization principle can be exploited to develop improved training techniques for deep learning.

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

# Supplementary Materials

## A   PROOF OF THEOREM 1

To prove item 1, note that the SA update rule implies that

$$
\begin{aligned}
\|\theta_{k+1} - \theta^*\|^2 &= \|\theta_k - \eta U(\theta_k; z_{\xi_k}) - \theta^*\|^2 \\
&= \|\theta_k - \theta^*\|^2 + \eta^2 \|U(\theta_k; z_{\xi_k})\|^2 - 2\eta\langle\theta_k - \theta^*, U(\theta_k; z_{\xi_k})\rangle \\
&\leq \|\theta_k - \theta^*\|^2 + \eta^2 \|U(\theta_k; z_{\xi_k})\|^2 - \eta^2 \|U(\theta_k; z_{\xi_k})\|^2 \\
&\quad - 2\eta\gamma\|\theta_0 - \theta^*\|^2 \big(\ell(\theta_k; z_{\xi_k}) - \ell(\theta^*; z_{\xi_k})\big),
\end{aligned}
\tag{4}
$$

where the last inequality follows from the $\gamma$-optimization principle in Definition 1. Note that $\theta^*$ is a common global minimizer for the loss on all indivisual data samples. Therefore, $\ell(\theta_k; z_{\xi_k}) - \ell(\theta^*; z_{\xi_k}) > 0$ for all $k$ and the above inequality further implies that

$$
\|\theta_{k+1} - \theta^*\|^2 \leq \|\theta_k - \theta^*\|^2,
$$

which completes the proof of item 1.

To prove item 2, note that the SA algorithm adopts the cyclic sampling with reshuffle scheme. Summing eq. (4) over the $B$-th epoch (i.e., $k = nB, nB + 1, ..., n(B + 1) - 1$) gives that

$$
\|\theta_{n(B+1)} - \theta^*\|^2 \leq \|\theta_{nB} - \theta^*\|^2 - 2\eta\gamma\|\theta_0 - \theta^*\|^2 \sum_{k=nB}^{n(B+1)-1} \big(\ell(\theta_k; z_{\xi_k}) - \ell(\theta^*; z_{\xi_k})\big).
$$

Further telescoping over the epoch index, we obtain that for all $B$

$$
\|\theta_{nB} - \theta^*\|^2 \leq \|\theta_0 - \theta^*\|^2 - 2\eta\gamma\|\theta_0 - \theta^*\|^2 \sum_{P=0}^{B-1} \sum_{k=nP}^{n(P+1)-1} \big(\ell(\theta_k; z_{\xi_k}) - \ell(\theta^*; z_{\xi_k})\big).
\tag{5}
$$

Note that the left hand side is nonnegative for all $B$ and $\ell(\theta_k; z_{\xi_k}) - \ell(\theta^*; z_{\xi_k}) \geq 0$ for all $k$. Due to the cyclic sampling with reshuffle scheme, every data sample is visited once in each epoch. Fix an $i \in \{1, ..., n\}$ and denote the sequence of iterations in which the data sample $z_i$ is sampled as $i(0), i(1), ..., i(B-1)$. We can rewrite the above inequality as

$$
\|\theta_{nB} - \theta^*\|^2 \leq \|\theta_0 - \theta^*\|^2 - 2\eta\gamma\|\theta_0 - \theta^*\|^2 \sum_{i=1}^{n} \sum_{P=0}^{B-1} \big(\ell(\theta_{i(P)}; z_i) - \ell(\theta^*; z_i)\big).
$$

Suppose for certain $i$ the sequence $\{\ell(\theta_{i(P)}; z_i)\}_P$ does not converge to the global minimum $\ell(\theta^*; z_i)$. Then, for any $\epsilon > 0$, $\ell(\theta_{i(P)}; z_i) - \ell(\theta^*; z_i) > \epsilon$ for infinitely many $P$. This implies that the above double summation diverges to $+\infty$ and the right hand side of the inequality eventually becomes negative, contradicting with the non-negativity of the left hand side. Therefore, $\ell(\theta_{i(P)}; z_i) \xrightarrow{P} \ell(\theta^*; z_i)$ for all $i$. This completes the proof of item 2.

Item 3 follows from eq. (5) after rearranging the terms and noting that $\theta^*$ is a common minimizer for all the indivisual loss functions.

## B   DISTANCE-TO-MINIMIZER RESULTS ON BATCH NORMALIZATION

All distances-to-minimizer diminishes monotonically and are consistent with the theoretical implications of the $\gamma$-optimization principle in items 1 of Theorem 1.

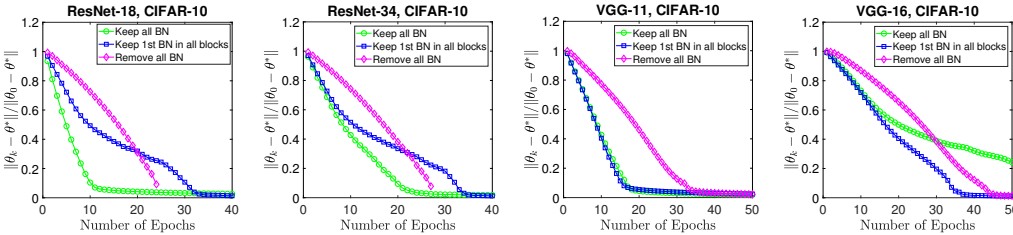

Figure 7: Training Resnets and Vggs with and without BN on Cifar10.

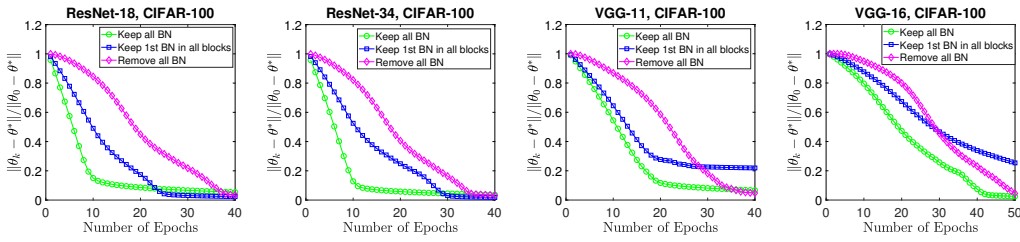

Figure 8: Training Resnets and Vggs with and without BN on Cifar100.

## C   DISTANCE-TO-MINIMIZER RESULTS ON SKIP CONNECTION

All distances-to-minimizer diminishes monotonically and are consistent with the theoretical implications of the $\gamma$-optimization principle in items 1 of Theorem 1.

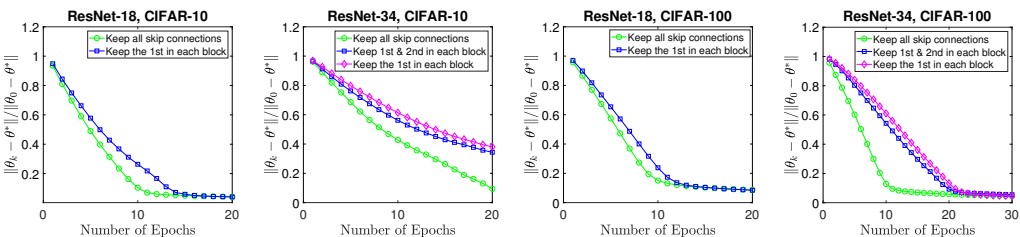

Figure 9: Training Resnets with and without skip connections.

# D MORE RESULTS ON OPTIMIZERS

## D.1 DISTANCE-TO-MINIMIZER RESULTS

All distances-to-minimizer diminishes monotonically and are consistent with the theoretical implications of the $\gamma$-optimization principle in items 1 of Theorem 1.

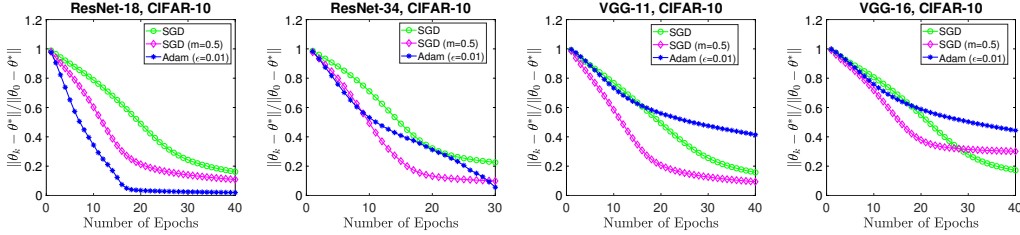

Figure 10: Training Resnets and Vggs with different optimizers on Cifar10.

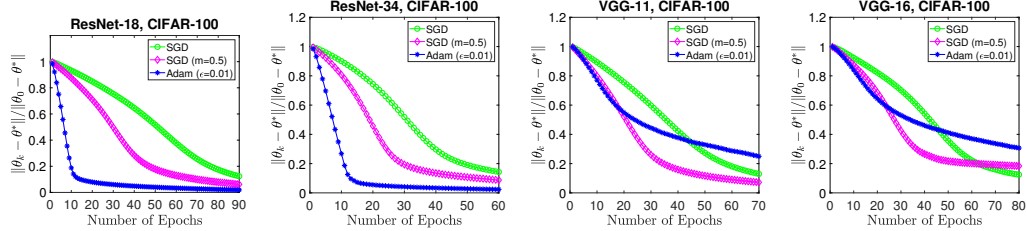

Figure 11: Training Resnets and Vggs with different optimizers on Cifar100.

## D.2 VISUALIZING OPTIMIZATION PATH OF DIFFERENT OPTIMIZERS

By exploiting the visualization method proposed in Li et al. (2018a), we plot the 2D visualization of the optimization paths of different optimizers on the loss landscape contours of the Resnet-18, 34, Vgg-11, 16 networks in Figure 12. In specific, we store the optimization paths generated by both the SGD with momentum and Adam ($\epsilon = 10^{-2}$). For the parameters in each optimization path, we compute their first and second principle components as the 2D axes. Moreover, we also calculate the SGD updates evaluated at the parameters along each optimization path and project their directions onto the two principle components for visualization.

From the visualization results shown in Figure 12, it can be seen that the directions of SGD updates are very different from those of the updates generated by the SGD with momentum and Adam. This is consistent with the experiments in the previous subsection where we observe that the $\gamma$ in the SGD trainings are very different from those in both the SGD with momentum trainings and Adam trainings.

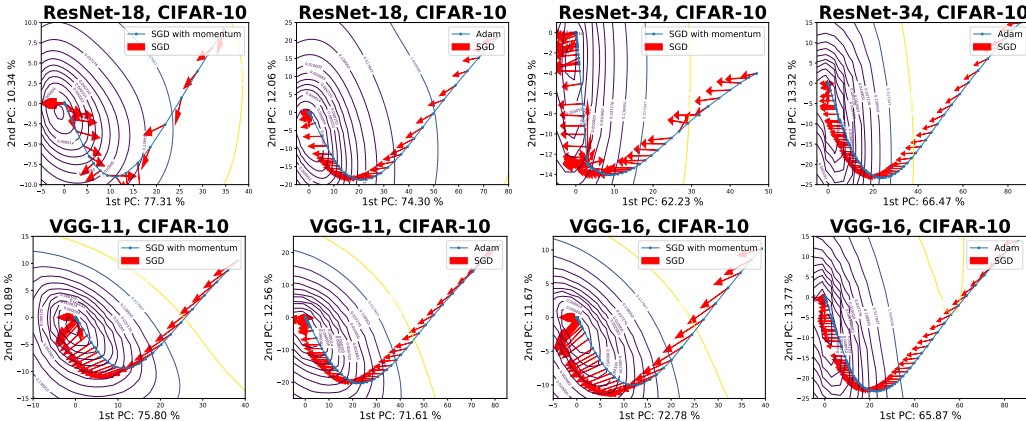

Figure 12: Visualization of optimization paths of different optimizers and SGD updates

