# OpenReview forum: "An Optimization Principle Of Deep Learning?"
_ICLR.cc/2020/Conference — Reject_

### Official Review · AnonReviewer1 · 2019-10-12
**Official Blind Review #1**

**Rating:** 1

**Review:**

Summary:

This paper proposes an optimization principle that is called \gamma-optimization principle for stochastic algorithms (SA) in nonconvex and over-parametrized optimization. The author(s) provide convergence results under this “\gamma-optimization principle” assumption. Numerical experiments are conducted on classification datasets CIFAR-10 and CIFAR-100 for Alexnet and Resnet-18 with different activation functions.

Comments:

1) Could you please explain how you could achieve the value of \theta* (common global minimizer for the loss on all individual component functions)? It is unclear to me how you could obtain it.

2) You have not mentioned the loss function that you are using for your numerical experiments. From my view, you are using softmax cross-entropy loss for classification problems (CIFAR-10, CIFAR-100). Can you show that Fact 1 is true for softmax cross-entropy loss? I wonder how you could train the total loss to achieve zero for this loss.

3) Fact 1 with over-parameterized model could be true if the loss, for example, is mean square (for regression problems). Therefore, I would suggest you to consider other numerical examples rather than classification problems. If not, the numerical part is not very consistent with the theoretical part.

4) The assumption that the author(s) use in the paper, that is, \gamma-optimization principle in Definition 1, is indeed strong and not reasonable. You simply assume what you want in order to achieve the convergence result. It is not easy to verify this assumption since you include \theta* unless it has only a unique solution. Note that the learning rate (eta) and gamma are very sensitive here and it is not clear how to determine these values.

5) There is related work that you may need to consider: Vaswani et al 2019, "Fast and Faster Convergence of SGD for Over-Parameterized Models (and an Accelerated Perceptron)" in AISTATS 2019.

I think the paper still needs lots of work to be ready. Theoretical result is not strong and the numerical experiments are not convincing. I do not support the publication for this paper at the current state.

Minor:
1) I am not really why you have a question mark (?) in the title.


**Experience Assessment:**

I have published in this field for several years.

**Review Assessment: Checking Correctness Of Derivations And Theory:**

I carefully checked the derivations and theory.

**Review Assessment: Checking Correctness Of Experiments:**

I carefully checked the experiments.

**Review Assessment: Thoroughness In Paper Reading:**

I read the paper thoroughly.

---

> ### Author Response · Authors · 2019-11-11
> **Response to Review #1**
>
> We thank the reviewer for providing valuable feedback. Below is our point-to-point response. Any further comment is very welcome.
>
> 1) Could you please explain ...
> A: We assume the (neural network) model has enough expressive power to interpolate all the training data samples. In our experiments, we used the non-negative cross-entropy loss function and train the total loss for a sufficient number of epochs to achieve a small value (1e-4). In this case, we empirically found that the model \theta produced in the last training iteration achieves small loss on all the data samples, and therefore we choose it as an approximation of the \theta*.
>
> 2) & 3) You have not mentioned the loss function that ...
> A: We thank the reviewer for providing valuable suggestions. We use the cross-entropy loss in the experiments. We understand that an exact minimum cannot be achieved for this loss and use the last training iteration to serve as an approximation. To fix this issue, we will adopt the MSE loss under the teacher-student setting to guarantee the existence of a known common global minimum (Please see our general response).
>
> 4) The assumption that the author(s) use in the paper...
> A: In general, one can propose many different optimization conditions that guarantee convergence in nonconvex optimization, e.g., the regularity condition in (Zhang et al. 2017b) for nonconvex phase retrieval, star-convex condition in (Zhou et al. 2019) for deep learning. However, these conditions cannot characterize the effects of the deep learning training techniques explored in this paper. What we empirically found is that the proposed \gamma-principle well characterizes these effects in terms of the value \gamma. Regarding the concern on the minimizer \theta*, we will adopt the teacher-student setting to bridge the gap between our theory and experiments.
>
> 5) There is related work that you may...
> A: We thank the reviewer for recommending this paper on stochastic optimization theory. We will cite  and discuss it in the related works.
>
> Minor: we will use a more informative title in the revision.

---

### Official Review · AnonReviewer2 · 2019-10-17
**Official Blind Review #2**

**Rating:** 1

**Review:**

The paper proposes a new condition: $\gamma$-optimization principle.
The principle states that the inner product between the difference between the current iterate and a fixed global optimum and the stochastic gradient (or more general) is larger than the squared the gradient norm, plus the product of squared norm of the difference between the initialization and the global minimum, and the loss of the current iterate.

Under this condition, the paper shows sublinear convergence.

Main Comments：
The proposed conditions are similar to many previous works, as pointed out by authors. With these kinds of conditions, proving global convergence is trivial.
One question is that the condition holds uniformly for all models and every sampled data point. There is no randomness in the condition. I would expect a condition that has some "randomness" in it, e.g., the condition holds in expectation over random sampling over the data.
The condition also requires a specific global minimizer. Because of the randomness in initialization and stochastic training, I expect the target global minimizer can change from iteration to iteration, but the current condition does not reflect that.


--------------------------------------------------------------------------------------------
I have read the rebuttal and I maintain the score.
Note in the student-teacher setting even though teacher is unique, there can be multiple optimal students. I don't think this resolves my concern.

**Experience Assessment:**

I have published in this field for several years.

**Review Assessment: Checking Correctness Of Derivations And Theory:**

I assessed the sensibility of the derivations and theory.

**Review Assessment: Checking Correctness Of Experiments:**

I assessed the sensibility of the experiments.

**Review Assessment: Thoroughness In Paper Reading:**

I read the paper at least twice and used my best judgement in assessing the paper.

---

> ### Author Response · Authors · 2019-11-11
> **Response to Review #2**
>
> We thank the reviewer for providing valuable feedback. Below is our point-to-point response. Any further comment is very welcome.
>
> 1) The proposed conditions are similar...
> A: We agree that the theoretical contribution of this work is not substantial. Our primary goal is to point out that the proposed simple optimization principle matches the optimization in practical deep learning and can characterize the effects of deep learning training techniques, and we further verify via extensive experiments.
>
> 2) One question is that the condition holds uniformly...
> A: Our experiments show that such a condition holds in practice for all the sampled data. We think this is because the network model is large enough to guarantee the existence of (approximate) common global minimizers, which encourages all loss functions to have a similar optimization path. In fact, one can propose the condition to be held in expectation over the random sampling and similar theoretical results can be derived. However, such a condition is very difficult to verify as the samples drawn within an epoch are not independent under cyclic sampling with reshuffle.
>
> 3) The condition also requires a specific global minimizer...
> A: We will fix this issue by using a teacher-student setting. Please see our general response for the details.

---

### Official Review · AnonReviewer3 · 2019-10-22
**Official Blind Review #3**

**Rating:** 3

**Review:**

This paper proposes a "gamma principle" for stochastic updates in deep neural network optimization. First, the authors propose if Eq. (2) is satisfied then convergence is guaranteed (Thm. 1). Second, they use experimental results of Alexnet and Resnet-18 on cifar10/100 to show that Eq. (2) is satisfied by SGD, SGD with momentum, and Adam, with different activations and tricks like batch normalization, skip connection.

Pros:
1. This paper is well written and the presentation is clear.
2. The experiments are extensive.

Cons:
1. Before Eq. (2), it is assumed that over-parameterized NNs are used as \theta. But there is no quantization how many parameters are enough? Are the Alexnet/Resnet-18 in experiments enough over-parameterized and how can we tell that? Some quantitative conditions should be provided to show what kind of models this principle hold.

2. The connection between Eq. (2) and Thm. 1 is too obvious and the gamma is just to characterize the progress of each update. Of course, large progress corresponds to fast convergence. Eq. (2) is a strong assumption rather than a theoretical contribution.

3. Experiments are used to show that Eq. (2) is a "principle". However, the experiments are problematic as follows.
First, "we set \theta^* to be the network parameters produced in the last training iteration", then how do we make sure \theta in the last training iteration is \theta^*, even if the loss is close to (but not exactly) zero? For this point, I suggest using a teacher-student setting, where the optimal \theta^* is already known.
Second, using \theta in the last training iteration makes the experiments show the following simple fact, that methods/tricks/activations with faster convergence to certain parameter will have larger every update progress to that parameter, which is, of course, true and does not reveal an optimization principle of deep learning.
Third, it is difficult to claim that this is a principle for general deep learning by using experiments on two datasets.

Overall, I found the theory not inspiring and experiments not convincing.

========Update=========
Thanks for the rebuttal.
I have read it and using teacher-student setting is an improvement to resolve my question with respect to \theta^*.
However I would maintain my rating since
1) the theoretical contribution is actually marginal;
2) the argument that this "gamma principle" holds for over-parameterized NNs is vague in the sense (and the author did not resolve my concern of this) that for what kinds of over-parameterized NNs this would work and for what kinds of NNs it does not hold. In particular, mentioning other theory work of over-parameterized NNs is not enough, because usually in these work, the numbers of parameters in NNs are poly(n), like O(n^4), O(n^6), where n is number of training data. There is an obvious gap between poly(n) and the number of parameters in experiments. From this perspective, the experiments cannot verify the claim that this "principle" holds for over-parameterized NNs that mentioned by authors in the rebuttal.
3) experiments on two datasets are not enough to claim this is a general "principle".
Considering this claim is for general over-parameterized NN optimization, I think it lacks of specifying types of NNs for which this claim would hold (of course it cannot hold for any NNs, but it possibly can hold for some NNs, and what are these NNs?), and experiments are not enough to show the generality of this claim.

**Experience Assessment:**

I have read many papers in this area.

**Review Assessment: Checking Correctness Of Derivations And Theory:**

I carefully checked the derivations and theory.

**Review Assessment: Checking Correctness Of Experiments:**

I carefully checked the experiments.

**Review Assessment: Thoroughness In Paper Reading:**

I read the paper thoroughly.

---

> ### Author Response · Authors · 2019-11-11
> **Response to Review #3**
>
> We thank the reviewer for providing valuable feedback. Below is our point-to-point response. Any further comment is very welcome.
>
> 1). Before Eq. (2), ...
> A: In our experiments, we found that all the neural network models can be trained to achieve 1e-4 total loss, which is very close to the global minimum. We think this justifies that these models are over-parameterized. On the other hand, we think a quantitative bound for over-parameterization can be obtained by following other theoretical works, e.g., Gradient Descent Finds Global Minima of Deep Neural Networks. We leave this part of development for future study.
>
> 2). The connection between Eq. (2) ...
> A: We agree that fast convergence must correspond to large progress. However, the key point here is to quantify such progress to characterize the effects of different deep learning training techniques. We want to show that the proposed simple principle can quantify these effects and match the empirical observations well.  We have pointed out in the paper that there are other existing optimization principles that guarantee convergence in nonconvex machine learning, but they do not match the experimental observations in training deep networks under various training techniques.
>
> 3). Experiments are used to show ...
> A: We thank the reviewer for providing valuable feedback. We appreciate the suggestion to use the teacher-student setting to guarantee the existence of a common global minimizer. Based on your suggestion, we plan to use a teacher-student setting in our experiments where the optimal \theta* is already known. Please find the details in our general response.
>
> 4) Second, using \theta ...
> A: We want to point out that \gamma along does not correspond to the per-step progress, which actually corresponds to the last term in eq(4) in the appendix. In that term, the loss gap, in general, varies in the optimization process and therefore the per-step progress is not a constant. Under the proposed principle, we showed that the \gamma fully determines the convergence speed and is verified in the experiments.
>
> 5) Third, it is difficult to claim ...
> A: We plan to do experiments on more large and complex datasets (e.g. ImageNet dataset).

---

### Author Response · Authors · 2019-11-11
**General Response**

We thank the reviewers for providing much valuable feedback. The reviewers are concerned about the use of the last iteration as an approximate common global minimizer in the current experiments. We plan to fix this issue by using a teacher-student setting as follows. We will train the model first using hard labels (the original labels) and obtain the model in the last training iteration, which is referred to as the teacher model. Then, we use the teacher model to get the soft labels (the probability mass after softmax) for all the training data samples. Then, we will train a student model (with the same architecture as the teacher model) from scratch on the training data with the soft labels. For this training, we will use either KL divergence or MSE loss, and it is clear that the teacher model is the common global minimizer for these loss functions on the modified data with soft labels. Lastly, we will verify the \gamma-optimization principle using the training path of the student model. We will also explore more datasets such as the imagenet.

---

### Decision · Program_Chairs · 2019-12-19

**Decision:**

Reject

**Comment:**

The paper is rejected based on unanimous reviews.